# Consensus designs and thermal stability determinants of a human glutamate transporter

**Erica Cirri[1,2], Sébastien Brier[3,4], Reda Assal[1,2], Juan Carlos Canul-Tec[1,2], Julia Chamot-Rooke[3,4], Nicolas Reyes[1,2]***

[1]Molecular Mechanisms of Membrane Transport Laboratory, Institut Pasteur, Paris, France; [2]UMR 3528, CNRS, Institut Pasteur, Paris, France; [3]Mass Spectrometry for Biology Unit, Institut Pasteur, Paris, France; [4]USR 2000, CNRS, Institut Pasteur, Paris, France

**Abstract** Human excitatory amino acid transporters (EAATs) take up the neurotransmitter glutamate in the brain and are essential to maintain excitatory neurotransmission. Our understanding of the EAATs' molecular mechanisms has been hampered by the lack of stability of purified protein samples for biophysical analyses. Here, we present approaches based on consensus mutagenesis to obtain thermostable EAAT1 variants that share up to ~95% amino acid identity with the wild type transporters, and remain natively folded and functional. Structural analyses of EAAT1 and the consensus designs using hydrogen-deuterium exchange linked to mass spectrometry show that small and highly cooperative unfolding events at the inter-subunit interface rate-limit their thermal denaturation, while the transport domain unfolds at a later stage in the unfolding pathway. Our findings provide structural insights into the kinetic stability of human glutamate transporters, and introduce general approaches to extend the lifetime of human membrane proteins for biophysical analyses.

DOI: https://doi.org/10.7554/eLife.40110.001

*For correspondence:
nreyes@pasteur.fr

## Introduction

Integral membrane proteins are essential to the life of any cell and constitute drug targets of paramount importance (*Yildirim et al., 2007*). The most established approach to obtain purified membrane proteins for molecular and structural studies involves their detergent solubilization and extraction from cell membranes (*Smith, 2017*). However, detergent micelles are rather poor surrogates of membranes, and tend to stabilize unfolded and dysfunctional states of membrane proteins (*Tate, 2010*). Current methods to overcome this problem involve mainly engineering stability through mutagenesis using amino acid scanning and directed evolution (*Scott et al., 2013*), and have been successfully used to stabilize both prokaryotic (*Zhou and Bowie, 2000*; *Faham et al., 2004*) and animal integral membrane proteins (*Magnani et al., 2008*; *Magnani et al., 2016*; *Sarkar et al., 2008*; *Scott and Plückthun, 2013*; *Coleman et al., 2016*). However, these approaches rely on screening of a large number of mutants and high-throughput methods to probe protein expression and stability, which makes them labor-intensive and sometimes impractical.

Methods based on phylogenetic analysis of amino acid sequences constitute an attractive alternative to predict stabilizing mutations (*Sauer et al., 2015*), in particular those using amino acid consensus from sequences of structural homologs (*Steipe, 2004*; *Porebski and Buckle, 2016*; *Lehmann and Wyss, 2001*). Pioneering work on antibody stability showed that consensus mutagenesis has much higher rate of success (~60%) than random mutagenesis (*Steipe et al., 1994*). Moreover, full consensus sequences of soluble proteins (*Lehmann et al., 2000*; *Sullivan et al., 2011*) or

repeated domains (*Mosavi et al., 2002*; *Kohl et al., 2003*) demonstrated significantly higher stability than the individual sequences that were used for their design, while combinatorial consensus mutagenesis was shown to improve the thermal stability of several soluble proteins (*Pantoliano et al., 1989*; *Ohage et al., 1997*; *Nikolova et al., 1998*). However, the applicability of these approaches to human membrane proteins in detergent solutions has remained unexplored.

Human glutamate transporters (EAATs) belong to the SoLute Carrier 1 (SLC1) family of ion-coupled transporters (*Slotboom et al., 1999*). Isoforms EAAT1 and EAAT2 are highly expressed in the central nervous system (*Danbolt, 2001*) where they take up glutamate using the energy stored in the sodium, proton and potassium transmembrane electrochemical gradients (*Zerangue and Kavanaugh, 1996*). These transporters and their mammalian orthologs are highly unstable in detergent solutions and loose their function during purification (*Gordon and Kanner, 1988*; *Shouffani and Kanner, 1990*), which has precluded the biophysical and structural characterization of purified transporters. Using consensus mutagenesis and loop engineering, we recently designed thermostable variants of human EAAT1 for crystallization (EAAT1$_{CRYST}$) and solved their 3D structures (*Canul-Tec et al., 2017*). Notably, these variants share ~75% amino acid sequence identity with the wild type transporter (EAAT1$_{WT}$) and cover ~95% of its sequence, constituting the closest structural homologs of human EAATs with known structure. EAAT1$_{CRYST}$ is a homo-trimer, in which each subunit contains two structural and functional domains, namely the scaffold (ScaD) and transport (TranD) domains, previously observed in the structures of a glutamate transporter prokaryotic homologue (*Yernool et al., 2004*; *Reyes et al., 2009*). The ScaD forms the inter-subunit interface through extensive contacts on both the extracellular half of the membrane between transmembrane helix 2 (TM2) and TM4a-b of contacting subunits, as well as on the cytoplasmic side between TM4c and TM5 of its neighboring subunit. Most of the remaining structured regions of the protein are alpha helical, including TM3 and TM6-8, as well as two re-entrant loops (HP1-2) that fold into the TranD that encages the substrate and the stoichiometrically coupled ions (*Boudker et al., 2007*; *Guskov et al., 2016*; *Seal and Amara, 1998*; *Tao et al., 2010*; *Zhang et al., 1998*). Substrate translocation across the membrane occurs through large rigid-body movements of the TranD relative to the ScaD in a so-called 'elevator-like' fashion (*Reyes et al., 2009*; *Crisman et al., 2009*) and shows no cooperativity between the subunits (*Ruan et al., 2017*).

In this work, we used consensus-based approaches to design EAAT1 variants that share as much as ~95% amino acid sequence identity, and remain folded and functional in detergent solutions. Structural comparison between folded and unfolded states of the transporters unravels key structural changes that determine the kinetic stability of the transporters, and provide unprecedented insights on their thermal denaturation pathway.

## Results

### EAAT1 consensus designs

EAAT1$_{WT}$ heterologously expresses at the cell surface of mammalian cells and is functional for neurotransmitter transport (*Arriza et al., 1994*). However, when detergent-purified and reconstituted into synthetic liposomes, EAAT1$_{WT}$ lacks this function suggesting that the detergent solutions used for solubilization and purification irreversibly inactivate the protein (*Canul-Tec et al., 2017*). To increase the stability of the transporters in detergent solutions, we identified the most frequent amino acids among representative animal SLC1 structural homologs (i.e. consensus amino acids; see methods), and simultaneously exchanged all residues within the expected helical regions of EAAT1$_{WT}$ for consensus amino acids. This consensus design yielded a transporter, so-called EAAT1-consensus (EAAT1$_{CO}$), with 44 and 33 amino acid exchanges in the ScaD and TranD, respectively (*Figure 1a*, *Figure 1—figure supplement 1*). The exchanges localize mostly to hydrophobic regions facing either the trimeric interface or the lipid bilayer and involve conservative substitutions among hydrophobic residues (mainly between Ile, Leu, Val, and Met). EAAT1$_{CO}$ shares 85% amino acid identity and 92% similarity with EAAT1$_{WT}$, and shows similar levels of neurotransmitter uptake in cells compared to EAAT1$_{WT}$. Unlike the wild-type protein however, EAAT1$_{CO}$ featured robust glutamate uptake upon purification and reconstitution in synthetic liposomes (*Figure 1c–e*). Moreover, glutamate transport in liposomes was strongly dependent on opposite gradients of sodium and potassium across the bilayer, and the rate of transport increased with the external L-glutamate

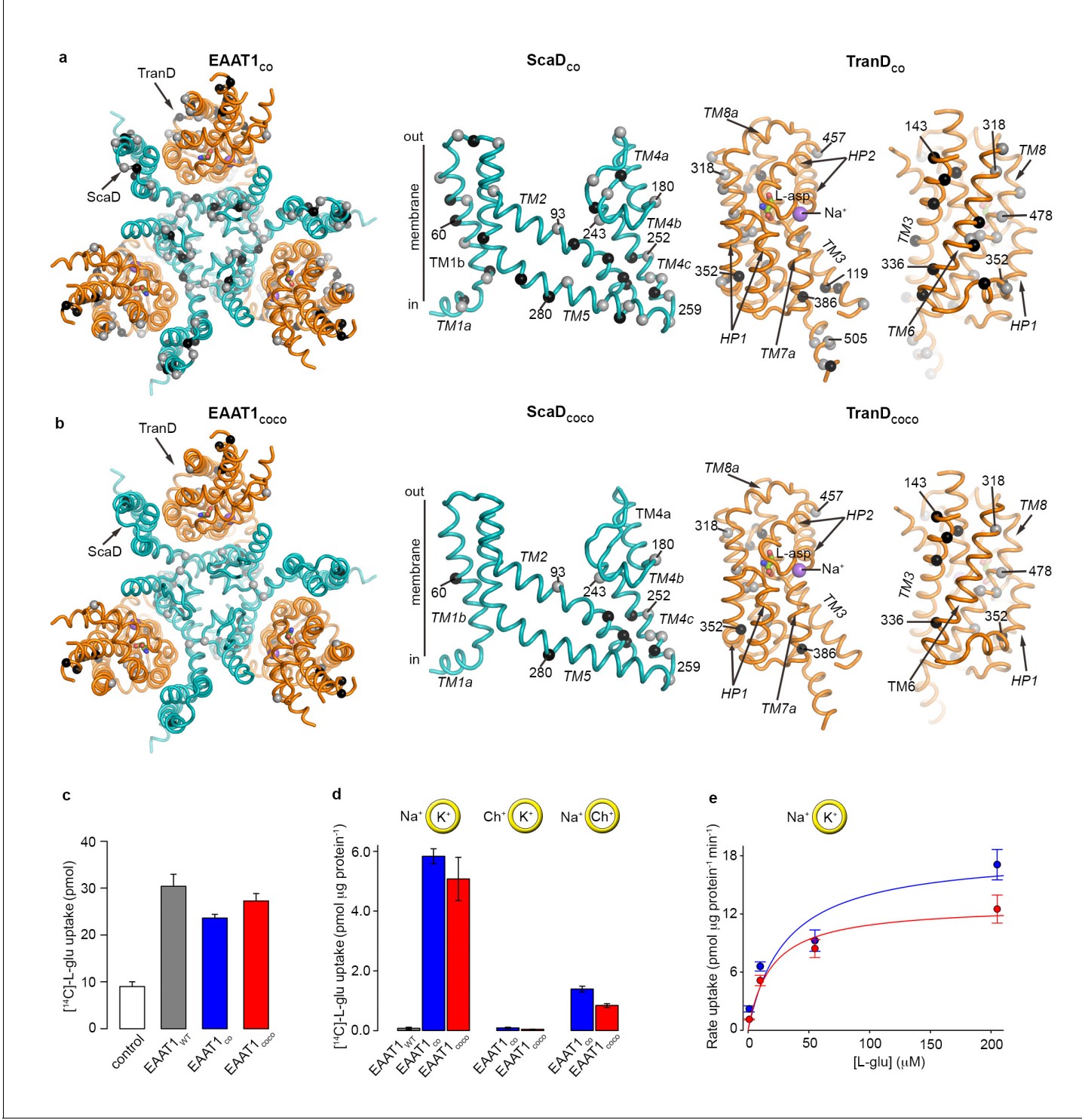

**Figure 1.** | EAAT1 consensus mutants. (a-b) Residues exchanged for consensus amino acids in EAAT1$_{CO}$ (a) and EAAT1$_{COCO}$ (b) are mapped into the structure of the EAAT1$_{CRYST}$ (PDB 5LLM) trimer viewed from the extracellular medium (left panel), as well as the scaffold (cyan) and the transport (orange) domains viewed from the membrane. These domains are depicted separately for clarity of display, including two views of the transport domain separated ~ 180° from each other, showing its interface with the scaffold domain (left) and the membrane (right), respectively. Spheres correspond to the alpha carbon atoms of residues that were exchanged by conservative (grey) and non-conservative (black) consensus mutations. (c–d), Radioactive L-glutamate uptake in HEK293 cells expressing the transporters (c), including control cells transfected with a vector lacking EAAT1 genes, and in liposomes with purified reconstituted transporters (d). EAAT1$_{WT}$ data in (d) was originally published in ref. (29). Yellow circles depict the liposomal bilayer separating sodium- (Na$^+$), potassium- (K$^+$), and choline-based (Ch$^+$) solutions. (e), Rate of L-glutamate uptake by purified EAAT1$_{CO}$ (blue) and

*Figure 1 continued on next page*

*Figure 1 continued*

EAAT1$_{COCO}$ (red) reconstituted in liposomes, as a function of L-glutamate concentration. Solid lines indicate Michaelis-Menten fits to the data with K$_m$ values 30.7 ± 25.6 and 18.8 ± 8.3 µM, and V$_{max}$ values 18.3 ± 4.3 and 12.8 ± 1.4 pmol µg$^{-1}$ min$^{-1}$ for EAAT1$_{CO}$ (blue) and EAAT1$_{COCO}$ (red), respectively. Plots in c–e) depict an average of at least three independent experiments performed with duplicate measurements, and error bars represent s.e.m.

DOI: https://doi.org/10.7554/eLife.40110.002

The following figure supplement is available for figure 1:

**Figure supplement 1.** | Amino acid alignment EAAT1 constructs.

DOI: https://doi.org/10.7554/eLife.40110.003

concentration with a K$_m$ (~30 µM) similar to the one reported for EAAT1$_{WT}$ (*Arriza et al., 1994*). Altogether, these data demonstrate that detergent-solubilized and purified EAAT1$_{CO}$ remains active and that its ion-coupled transport mechanism is conserved.

In a second consensus design, we hypothesized that coevolved residue-residue contacts could play an important role in protein stability. Amino acid covariance in large sequence alignments has been shown to accurately predict amino acid physical proximity in 3D protein structures (*Morcos et al., 2011*; *Hopf et al., 2012*; *Kamisetty et al., 2013*; *Nugent and Jones, 2012*), and we used it as a proxy for coevolved inter-residue contacts. Amino acid covariance was calculated from a curated PFAM alignment of the SLC1 family, and the consensus exchanges were restricted to positions in the EAAT1$_{WT}$ sequence with the highest covariance (see methods). It is worth noting that the consensus amino acid exchanges were identical to those in EAAT1$_{CO}$, and determined from animal SLC1 homologs. This alternative design yielded a transporter, the so-called EAAT1-consensus-covariance (EAAT1$_{COCO}$), that contains a subset of the EAAT1$_{CO}$ amino acid exchanges, 15 in the ScaD and 14 in TranD, and shares high amino acid sequence identity (95%) and similarity (97%) with EAAT1$_{WT}$ (*Figure 1b*, *Figure 1—figure supplement 1*). Like EAAT1$_{CO}$, purified EAAT1$_{COCO}$ retained conserved ion-coupled transport mechanism upon reconstitution in synthetic liposomes (*Figure 1c–e*).

## Structural comparison between EAAT1 wild type and consensus designs

To gain insights into the structural differences between EAAT1$_{WT}$ and the consensus designs in detergent solutions, we compared their hydrogen-deuterium exchange (HDX) behavior using a protease-induced fragmentation approach linked to mass spectrometry (MS) (*Konermann et al., 2011*; *Englander et al., 2016*). HDX-MS measures the rate of deuterium exchange of the hydrogen amide in the protein backbone that strongly depends on the presence of secondary structure, due to the engagement of the amide group in hydrogen bonding, as well as on its access to the aqueous solvent. Therefore, HDX-MS provides very valuable information on protein folding, stability, and dynamics.

EAAT1$_{CO}$ and EAAT1$_{COCO}$ showed similar overall HDX patterns when assayed at 20°C in detergent solutions with overall sequence coverage of ~70% (*Figure 2a–e*, *Figure 2—figure supplements 1* and *2*), and unimodal isotopic envelopes across all peptides (*Figure 2*, *Figure 2—figure supplement 3*). The deuterium uptake time course of most peptides covering structured regions of the TranD and ScaD based on the structure of EAAT1$_{CRYST}$ showed slow deuterium incorporation that starts after ≥10 s, or even lacked any incorporation for up to 1 hr (*Figure 2a–c*, *Figure 2—figure supplement 4*). The observed kinetics is several orders of magnitude slower than predicted for unstructured peptides that are expected to reach saturating uptake values in the millisecond range under our experimental conditions (*Bai et al., 1993*). This demonstrates high degree of backbone protection of those peptides and strongly suggests conservation of the overall secondary structure in the transporters. Notably, TranD peptides covering buried regions that lack secondary structure and are involved in coordination of substrate and sodium ions, like the SLC1-family signature sequence 398-NMDG-401, also showed high degree of HDX backbone protection arguing that ligands remain bound and occluded from the bulk solution as observed in the crystal structures. In contrast, peptides covering predicted unstructured and non-conserved regions like the N- and C-termini, as well as the extracellular loops connecting TM3-TM4a and TM4b-TM4c, respectively, which were not resolved in the EAAT1$_{CRYST}$ structures, showed saturating levels of deuterium uptake

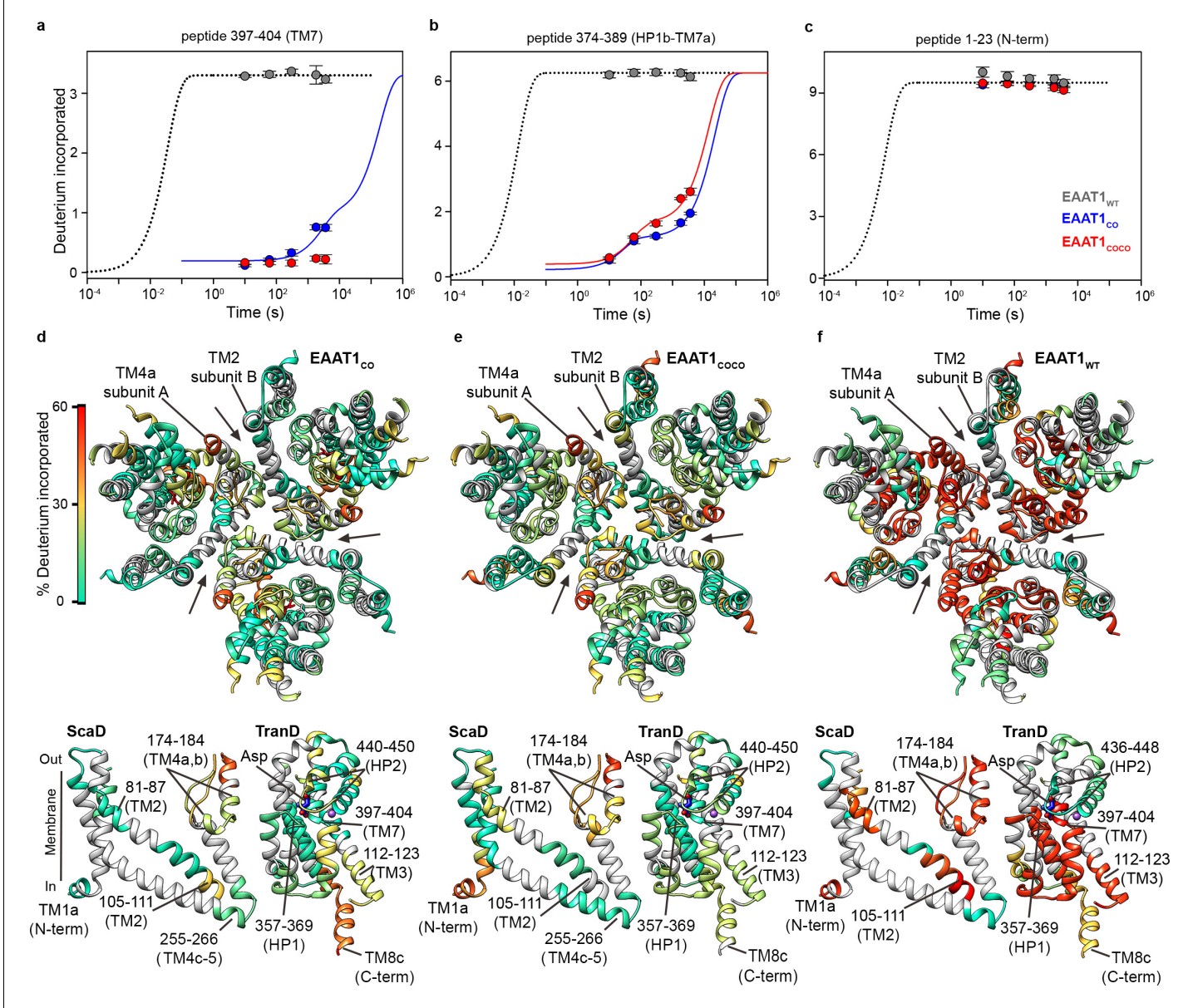

**Figure 2.** | Deuterium exchange at 20°C. (**a-c**) Deuterium uptake kinetics at 20°C of examples peptides from EAAT1$_{WT}$ (grey), EAAT1$_{CO}$ (blue), and EAAT1$_{COCO}$ (red), respectively, covering both helical and unstructured regions of the transporters. Solid lines represent double-exponential fits to the data, and dotted lines the expected deuterium kinetics of unfolded and solvent exposed peptides (see 'HDX kinetic analysis' in Methods). Plots in (**a–c**) depict an average of three independent experiments, and error bars represent s.e.m. (**d-f**) Deuterium incorporation after 1 hr at 20°C in EAAT1$_{CO}$ (**d**), EAAT1$_{COCO}$ (**e**), and EAAT1$_{WT}$ (**f**) mapped into the structure of the EAAT1$_{CRYTS}$ (PDB 5LLM) trimer viewed from the extracellular medium (upper panel), as well as the scaffold (ScaD) and the transport (TranD) domains viewed from the membrane (lower panel), respectively. These domains are depicted separately for clarity of display, and black lines indicate the approximate position of some peptides in the structure, and the substrate (Asp). In the trimeric depiction, arrows point to the interface between protomers. Deuterium incorporation was calculated as an average of three independent experiments, and normalized to the maximal theoretical incorporation based on the number of backbone amide available for exchange in each peptide. The color code representing deuterium incorporation is depicted in a scale bar (**d**).
DOI: https://doi.org/10.7554/eLife.40110.004

The following figure supplements are available for figure 2:

**Figure supplement 1.** | Deuterium exchange behavior of EAAT1$_{CO}$.
DOI: https://doi.org/10.7554/eLife.40110.005

**Figure supplement 2.** | Deuterium exchange behavior of EATT$_{COCO}$.
DOI: https://doi.org/10.7554/eLife.40110.006

*Figure 2 continued on next page*

*Figure 2 continued*

**Figure supplement 3.** | Examples of representative unimodal *m/z* envelopes (EX2 kinetics) observed in EAAT1$_{CO}$ and EAAT1$_{COCO}$.
DOI: https://doi.org/10.7554/eLife.40110.007
**Figure supplement 4.** | Deuterium uptake kinetics in EAAT1$_{WT}$, EAAT1$_{CO}$, and EAAT1$_{COCO}$.
DOI: https://doi.org/10.7554/eLife.40110.008
**Figure supplement 5.** | Deuterium exchange behavior of EAAT1$_{WT}$.
DOI: https://doi.org/10.7554/eLife.40110.009

already at the shortest time measured of 10 s, consistent with these regions being fully solvent accessible. Importantly, the HDX patterns of EAAT1$_{CO}$ and EAAT1$_{COCO}$ are similar to that of EAAT1-$_{CRYST}$ (*Canul-Tec et al., 2017*), which is also a functional transporter, suggesting that the overall secondary structure is similar among these transporters.

The HDX behavior of detergent-solubilized EAAT1$_{WT}$ was strikingly different from the consensus designs and showed an overall dramatic increase in deuterium uptake over most of the sequence coverage (*Figure 2a–c and f*, *Figure 2—figure supplements 4* and *5*). The HDX increase was most prominent in the cytoplasmic half of the TranD including TM3 (peptide 112 – 123) and importantly, in peptides covering the substrate and sodium binding sites in HP1 (peptide 357 – 369), TM7 (peptides 390 – 399 and 397 – 404), and TM8b (peptide 479 – 492). These regions of the TranD revealed maximal deuterium uptake already at 10 s, just as we observed in the unstructured N- and C-termini. The dramatic loss of HDX backbone protection at the functional core of the wild type transporter, compared to the consensus designs, implies that its native structure along with the interactions with the substrate and sodium ions was lost upon detergent-solubilization and purification. We also observed increased HDX in the ScaD, particularly in extracellular TM4a-b (including peptide 174 – 184, and 187 – 194) that forms extensive inter-subunit contacts, as well as the cytoplasmic end of TM2 (peptide 98 – 104) connecting the ScaD to the TranD at the level of TM3. Although the deuterium uptake time course in these regions still reveals some level of backbone protection, the large increase in deuterium uptake argues that the ScaD in EAAT1$_{WT}$ has partly lost secondary structure and/or that the oligomeric state of the transporter is compromised.

Altogether the HDX results show that EAAT1$_{CO}$ and EAAT1$_{COCO}$ retain the native fold observe in the structure of EAAT1$_{CRYST}$, while EAAT1$_{WT}$ undergoes partial and extensive unfolding events that include the substrate and coupled-sodium binding sites. Consistently, purified EAAT1$_{CO}$ and EAAT1$_{COCO}$, but not EAAT1$_{WT}$ showed robust transport function. The above results prove that the consensus, as well as the consensus-covariance designs can be used as semi-rational approaches to stabilize human EAAT1 and possibly other membrane proteins for biophysical analyses of their molecular mechanisms.

## Thermal stability of trimeric transporters

To gain further insights into the stability of EAAT1$_{CO}$ and EAAT1$_{COCO}$, we studied their thermal denaturation. Thermal denaturation is in general an irreversible process in detergent-solubilized membrane proteins, and offers the possibility to capture kinetic intermediates of the unfolding pathway. We first used size exclusion chromatography (SEC) to analyze the effect of single twenty-minute temperature pre-pulses on purified EAAT1$_{CO}$ and EAAT1$_{COCO}$ in detergent solutions, at a constant protein concentration. In this way, we aimed to quantify the irreversible effect of temperature on protein solubility, aggregation, and oligomeric state compared to reference transporters that were not pre-heated.

Indeed, both EAAT1$_{CO}$ and EAAT1$_{COCO}$ reference samples maintained at 4°C eluted as monodisperse peaks that correspond to the trimeric form of the transporters, and remained stable up to 35°C and 25°C, respectively (*Figure 3a,b,d,e*). At higher temperatures the trimers unfolded into a lower oligomeric state accompanied by a significant right-shift in the elution volume, but without any sign of aggregation or insolubility judging by the lack of high molecular-weight peaks and the constant area under the chromatograms at all temperatures, respectively. The temperature at which half of the transporters were in the trimeric state (T$_{50-SEC}$) in EAAT1$_{CO}$ and EAAT1$_{COCO}$ were 49.06 ± 0.02°C and 38.4 ± 0.3°C, respectively, demonstrating that the quaternary structure of EAAT1$_{CO}$ is kinetically more thermo-stable than that of EAAT1$_{COCO}$. Interestingly, the appearance of a single low molecular-weight peak in the chromatographic profiles at high temperatures argues that the

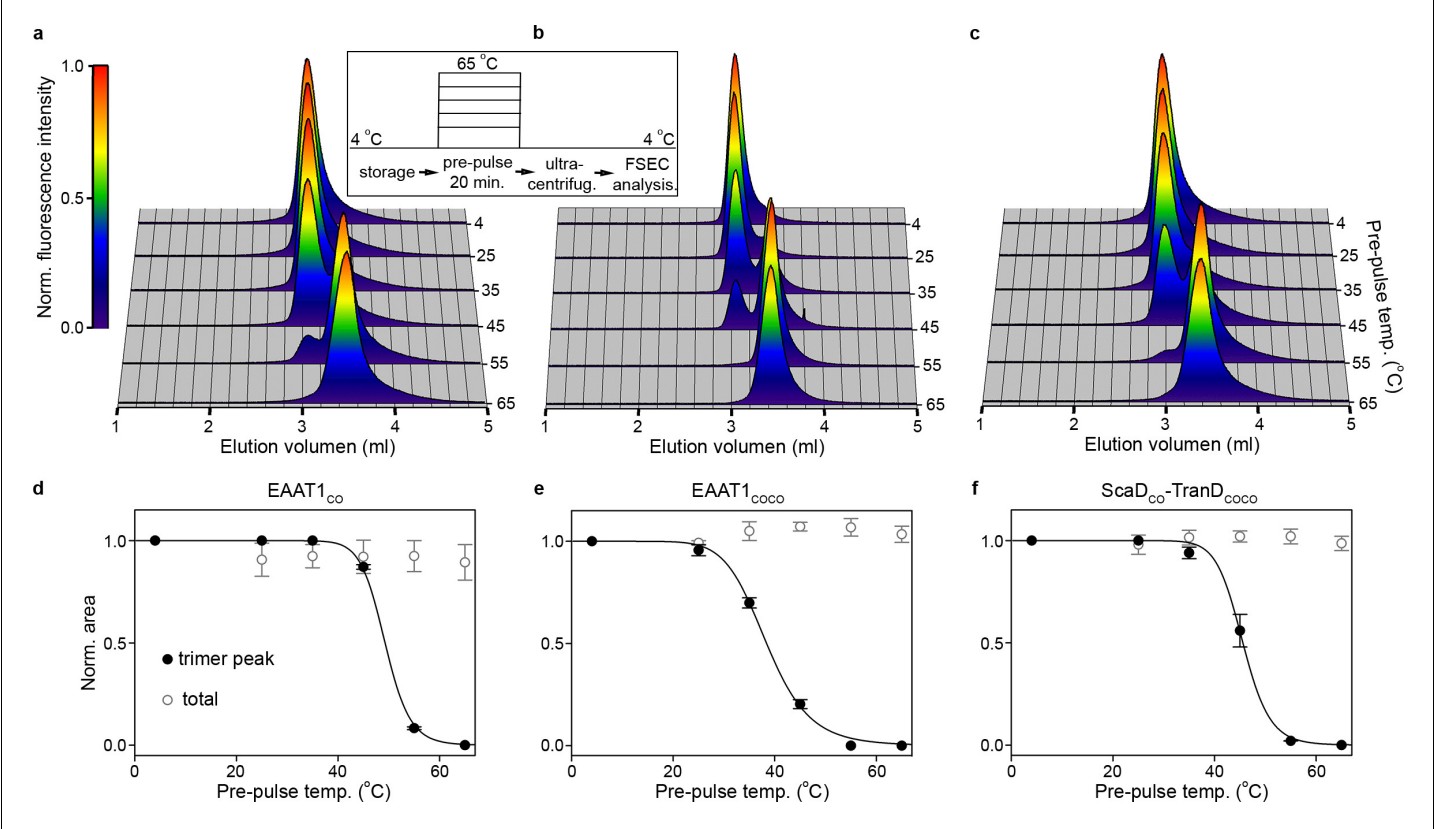

**Figure 3.** | Subunit dissociation by SEC. (**a-c**) Size-exclusion chromatograms of purified EAAT1$_{CO}$ (**a**), EAAT1$_{COCO}$ (**b**), and the chimeric transporter EAAT1-ScaD$_{CO}$-TranD$_{COCO}$ (**c**), respectively, pre-heated at different temperatures. The chromatograms at all temperatures were normalized to the peak fluorescence value observed at 4°C, and show how the trimeric form of the transporters that elutes at ~3.0 ml unfolds into lower oligomeric state(s), most likely monomers that elute at ~3.5 ml. (**d-f**) Thermal unfolding curves depicting the change in fractional area of the chromatographic peak corresponding to the trimeric transporters (black symbols), as a function of the pre-pulse temperature in EAAT1$_{CO}$ (**d**), EAAT1$_{COCO}$ (**e**), and EAAT1-ScaD$_{CO}$-TranD$_{COCO}$ (**f**), respectively. Solid lines indicate fits of a Hill-like equation (see methods) to the data with T$_{50\text{-SEC}}$ values 49.06 ± 0.1, 38.4 ± 0.4°C and 45.6 ± 0.4°C, and H values −21.0 ± 0.3, −9.3 ± 0.7, and −18.3 ± 5.3 for EAAT1$_{CO}$, EAAT1$_{COCO}$, and EAAT1-ScaD$_{CO}$-TranD$_{COCO}$, respectively. The total area under the chromatogram at each temperature, normalized to that at 4°C, is also shown (empty symbols), and remains relatively constant at all temperatures showing the lack of protein aggregation or loss during sample preparation. Plots in (**d**), (**e**), and (**f**) depict an average of at least three independent experiments (circles), and error bars represent s.e.m.

DOI: https://doi.org/10.7554/eLife.40110.010

The following figure supplement is available for figure 3:

**Figure supplement 1.** | Neurotransmitter uptake by EAAT1$_{COCO}$ pre-heated at 55°C.
DOI: https://doi.org/10.7554/eLife.40110.011

trimers unfold into monomers, in a highly cooperative process that does not involve the formation of stable dimers, and that monomers remain soluble at pre-pulse temperatures up to 65°C.

In order to probe the transport function of the low molecular-weight species observed by SEC, we reconstituted EAAT1$_{COCO}$ samples pre-heated at 55°C in liposomes, and compared them to the reference samples maintained at 4°C. Indeed, the large decrease in neurotransmitter uptake observed in the pre-heated samples shows that the transport function of the low molecular-weight species has been largely impaired (***Figure 3—figure supplement 1***), and indicates that the monomeric form of the transporter is not functional.

The concomitant loss of trimeric state and transport function argues that early events during the thermal inactivation of the transporters involve the cooperative dissociation of the subunits, and suggests an important role of the trimeric interface in the thermal stability of the transporters. To gain further insights into the role of this interface, we built a transporter with the ScaD of EAAT1$_{CO}$ and the TranD of EAAT1$_{COCO}$ (EAAT1-ScaD$_{CO}$-TranD$_{COCO}$), which involved adding 29 consensus

mutations into the EAAT1$_{COCO}$ ScaD (*Figure 1*, *Figure 1—figure supplement 1*). Notably, the temperature unfolding curve of EAAT1-ScaD$_{CO}$-TranD$_{COCO}$ approached that of EAAT1$_{CO}$ (T$_{50\text{-SEC}}$ of 45.2 ± 1.0°C; *Figure 3f*) confirming that key determinants of the transporter thermal stability localize to the trimeric interface.

## Local thermal unfolding

The chromatographic and functional analysis of the consensus designs clearly established that temperature pre-pulses irreversibly generate soluble intermediates of the thermal unfolding pathway. To gain structural insights into these intermediates, we measured deuterium uptake at 20°C in transporters pre-heated at different temperatures, using as reference transporters that were not pre-heated. Since the sample throughput by HDX-MS is limited, we focused the analysis of EAAT1$_{CO}$ and EAAT1$_{COCO}$ to temperatures corresponding to their respective T$_{50\text{-SEC}}$, as well as selected temperatures below and above it.

The overall HDX behavior of EAAT1$_{CO}$ and EAAT1$_{COCO}$ was similar when the transporters were pre-heated at temperatures close to their T$_{50\text{-SEC}}$, 50°C and 40°C, respectively (*Figure 4a,b*; *Figure 4—figure supplements 1a,b* and *2a*). In the TranD of the two consensus designs, deuterium uptake remained unchanged showing that within the limits of our sequence coverage, there were no irreversible structural changes at temperatures near the T$_{50\text{-SEC}}$. Strikingly, the two consensus designs showed significant HDX increases in the extracellular part of TM4 (e.g. peptide 174 – 184) and TM2 (e.g. peptide 81 – 87), which localize to the inter-subunit interface in the trimeric transporters. To a lesser extent, HDX also increased in the cytoplasmic parts of TM2 and TM3 in EAAT1$_{CO}$ (e.g. peptides 105–111 and 112–123, respectively), as well as the extracellular end of TM4c in EAAT1$_{COCO}$ (peptide 230–246), which are in close proximity to that interface (*Figure 4a,b*). These results map the initial temperature-induced structural changes in the transporters to the extracellular part of the inter-subunit interface, and reinforce its key role in thermal stability.

Next, we measured HDX in EAAT1$_{CO}$ and EAAT1$_{COCO}$ pre-heated at 65°C and 55°C (*Figure 4—figure supplements 1c* and *2b*), respectively. Both transporters showed dramatic HDX increases at the inter-subunit interface and its proximity (e.g. TM4 peptide 174–184; and TM2 peptide 81–87), which went beyond the changes observed at their respective T$_{50\text{-SEC}}$. Such extensive loss of backbone protection is consistent with the subunit dissociation observed by SEC that would expose the buried inter-subunit interface to the solvent. Furthermore, the fact that deuterium uptake at high temperature pre-pulses reaches nearly saturating values at 10–60 s in TM4a-c peptides 162–172 and 187–194 (*Figure 6—figure supplement 1*, cyan circles), strongly argues that there is also loss of secondary structure at the interfacial helices.

On the other hand, the HDX profile of the TranD differed greatly between EAAT1$_{CO}$ and EAAT1$_{COCO}$ at temperatures above their T$_{50\text{-SEC}}$. In the latter, deuterium uptake remained unchanged in samples preheated at 55°C, demonstrating lack of significant structural changes (*Figure 4—figure supplement 2b*). In contrast, EAAT1$_{CO}$ preheated at 65°C showed an overall dramatic HDX increase in the TranD (e.g peptide 361–369 in HP1, 390–399 in TM7, and 483–495 in TM8) (*Figure 4—figure supplement 1c*) consistent with extensive unfolding of this domain, including the substrate and sodium binding sites, just as we observed in EAAT1$_{WT}$. All together, these results show that the TranD of both EAAT1$_{CO}$ and EAAT1$_{COCO}$ have higher thermal stability than the respective ScaD, and remain folded at pre-pulse temperatures < 50°C.

## Correlation between local unfolding and subunit dissociation

Our thermal denaturation experiments show that increasing temperatures induce on one side loss of the trimeric state as monitored by SEC, as well as local unfolding that initially map to the inter-subunit interface, as indicated by HDX. Both types of structural changes are irreversible in nature and reflect the kinetic stability of the transporters, rather than their thermodynamic stability. Hence, the comparison between the T$_{50}$ values for the subunit dissociation (T$_{50\text{-SEC}}$) and those for the unfolding of individual peptides based on HDX (T$_{50\text{-HDX}}$), offers a quantitative way to correlate the temporal course of the two processes.

With that aim in mind, we observed that peptides at the inter-subunit interface (174 – 184 and 187 – 194, TM4a-b) in both EAAT1$_{CO}$ and EAAT1$_{COCO}$ showed clear bimodal isotopic envelops in the mass spectra reflecting two populations of transporters with significantly different HDX

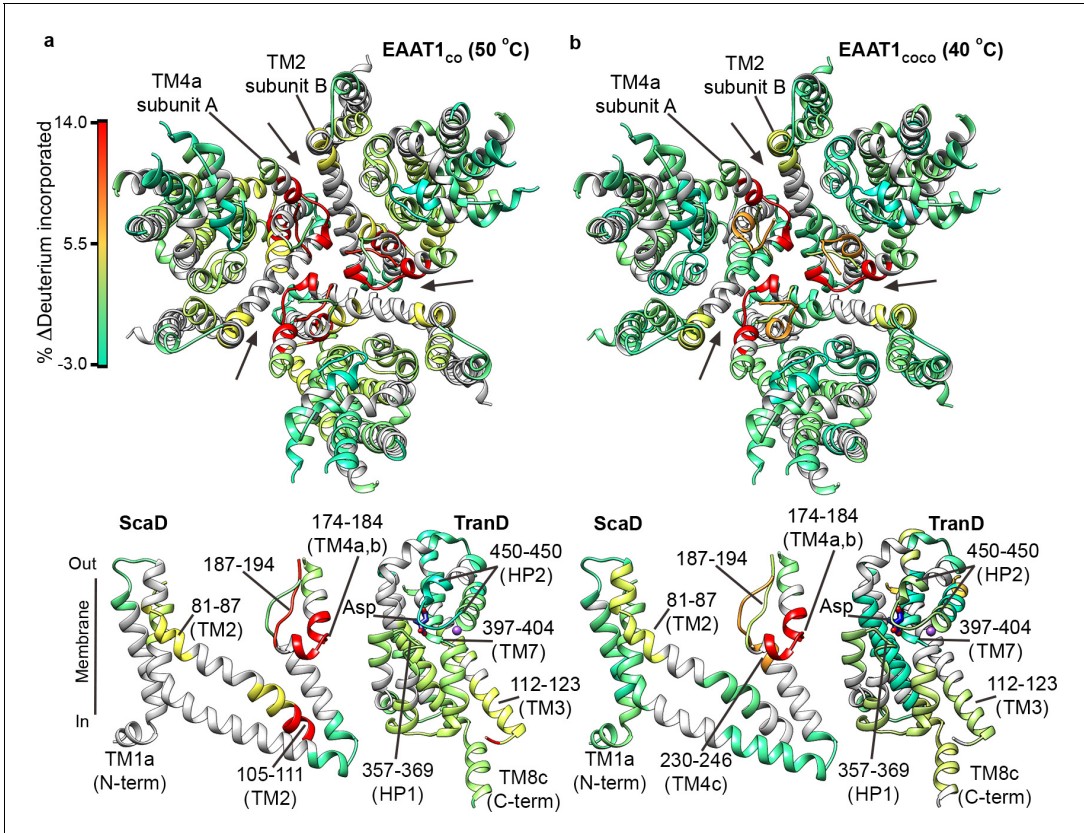

**Figure 4.** | Temperature-induced HDX changes. (a-b) Changes in deuterium incorporation induced by pre-pulses at nearly the $T_{50\text{-SEC}}$ of EAAT1$_{CO}$ (50°C) (a), and EAAT1$_{COCO}$ (40°C) (b), respectively, measured after 1 hr at 20°C. The changes are mapped into the structure of the EAAT1$_{CRYST}$ (PDB 5LLM) trimer viewed from the extracellular medium (upper panel), as well as the scaffold (ScaD) and the transport (TranD) domains viewed from the membrane (lower panel), respectively. These domains are depicted separately for clarity of display, and black lines indicate the approximate position of some peptides in the structure, and the substrate (Asp). In the trimeric depiction, arrows point to the interface between protomers. The color code representing the change in deuterium incorporation is depicted in a scale bar (a). Positive values represent increase in deuterium uptake at 50°C (EAAT1$_{CO}$) and 40°C (EAAT1$_{COCO}$) over the reference temperature (20°C), respectively.

DOI: https://doi.org/10.7554/eLife.40110.012

The following figure supplements are available for figure 4:

**Figure supplement 1.** | Temperature-induced changes in EAAT1$_{CO}$ HDX pattern.
DOI: https://doi.org/10.7554/eLife.40110.013

**Figure supplement 2.** | Temperature-induced changes in EAAT1$_{COCO}$ HDX pattern.
DOI: https://doi.org/10.7554/eLife.40110.014

dynamics. All other peptides were unimodal. Notably, the bimodal envelopes in TM4a-b were only present at pre-pulse temperatures close to the $T_{50\text{-SEC}}$, in which the amplitudes of the high- and low-mass components were similar (*Figure 5a,b*). In contrast, at temperatures above and below $T_{50\text{-SEC}}$ the envelopes were nearly unimodal and dominated by the high- and low-mass components, respectively. Importantly, the HDX time course of the low-mass component at the $T_{50\text{-SEC}}$ was nearly identical to that of the reference condition (20°C) (*Figure 5c,d*), indicating that this component arises from trimeric and natively folded transporters. On the other hand, the HDX time course of the high-mass component at the $T_{50\text{-SEC}}$ resembled that observed at high pre-pulse temperatures (65°C and 55°C in EAAT1$_{CO}$ and EAAT1$_{COCO}$, respectively), indicating that the high-mass component originates from denatured monomeric transporters in which TM4a-b are locally and partly unfolded. Therefore, we computed the temperature-induced changes in the amplitude of the low-mass component to estimate the $T_{50}$ of interfacial peptides ($T_{50\text{-HDX-Bi}}$) using Gaussian fitting. Peptides 174–184 and 187–194 in EAAT1$_{CO}$ showed $T_{50\text{-HDX-Bi}}$ values of 49.5°C and 48.8°C, respectively, while those in EAAT1$_{COCO}$ were 36.8°C and 42.4°C, respectively. The excellent agreement between $T_{50\text{-HDX-Bi}}$ and

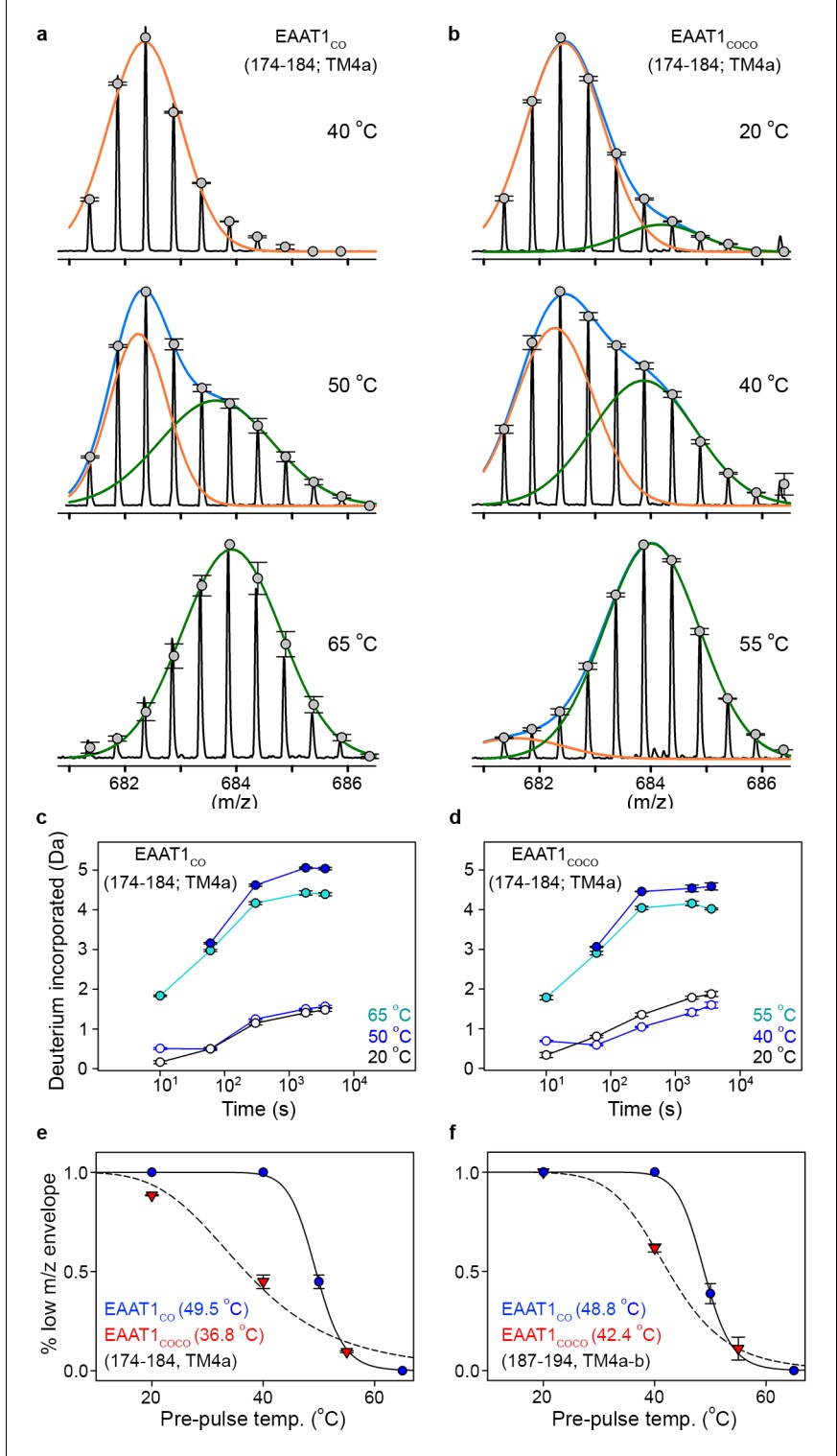

**Figure 5.** | Temperature-induced bimodal m/z envelopes at the trimeric interface. (a-b) m/z envelopes of an example peptide covering residues 174–184 of EAAT1$_{CO}$ (a) and EAAT1$_{COCO}$ (b), respectively, at different pre-pulse temperatures. Grey symbols represent the average of three experiments and error bars represent s.e.m, and are superimposed on the m/z spectrum of a representative experiment (black solid line). Blue solid lines represent fits of a double Gaussian equation to the data. The low-m/z (orange solid line) and high-m/z (green solid lines) components of such double-Gaussian fits are also shown. (c–d), HDX temporal course of peptide 174–184 of EAAT1$_{CO}$ (c) and EAAT1$_{COCO}$ (d) samples pre-heated at different temperatures. Deuterium incorporation values

*Figure 5 continued on next page*

*Figure 5 continued*

were calculated for the low-mass (blue open circles) and high-mass (blue solid circles) components of bimodal isotopic envelopes at the $T_{50\text{-SEC}}$, as well as unimodal envelopes at high (cyan solid circles) and reference (black open circles) temperatures. Plots are an average of three independent experiments, and error bars represent s. e.m. (e–f), Pre-pulse temperature dependence of the low-m/z component in peptides 174–184 (**e**) and 187–194 (**f**) of EAAT1$_{CO}$ (blue), and EAAT1$_{COCO}$ (red). Solid and dashed lines indicate fits of a Hill-like equation to the data with $T_{50\text{-HDX-Bi}}$ values 49.5 ± 0.1°C and 36.8 ± 2°C, and H values −20 ± 3.1 and −4.4 ± 1 for peptide 174–184 in EAAT1$_{CO}$ and EAAT1$_{COCO}$, respectively, as well as $T_{50\text{-HDX-Bi}}$ values 48.8 ± 0.2°C and 42.4 ± 0.1°C, and H values −20 ± 3.2 and −8 ± 0.1 for peptide 187–194 in EAAT1$_{CO}$ and EAAT1$_{COCO}$, respectively. Plots depict an average of amplitudes measured at least at four different time points in three independent experiments, and error bars represent s.e.m.

DOI: https://doi.org/10.7554/eLife.40110.015

$T_{50\text{-SEC}}$ values shows that subunit dissociation and unfolding of the inter-subunit interface are highly correlated and simultaneous processes.

## Rate-limiting structural changes of thermal unfolding

The above analyses can only be applied to peptides with bimodal isotopic envelopes, precluding a more extensive comparison of the local unfolding kinetics in other regions of the transporters. To overcome this problem, we calculated $T_{50}$ values of individual peptides from the uptake kinetics of unimodal isotopic envelopes ($T_{50\text{-HDX-Uni}}$). Briefly, the HDX kinetics of any detected peptide was well described by three components: an initial burst determined by uptake measured at 10 s (the first time point in the experiments); an intermediate component determined by uptake within $10^1$–$10^3$ s (where most of our experimental measurements were done); and a slow component with uptake at longer times > $10^3$ s. Despite the scarcity of experimental data covering this temporal window, with our longest uptake point measured at 3600 s, the slow component is required to reach saturating uptake values at pre-pulses ≤ 50°C (***Figure 6a–c***). The HDX kinetics of the structure regions of reference transporter samples (20°C) was dominated by the slow component, while the amplitude of the initial burst was nearly zero (***Figure 6a,b***; ***Figure 6—figure supplement 1***). At increasing pre-pulse temperatures, the slow component gradually disappeared and instead, the HDX kinetics was dominated by the burst and/or the intermediate components reflecting loss of backbone protection. Therefore, we computed the temperature-induced changes of the slow component amplitude as a proxy for the loss of native folded state, and extracted the associated $T_{50\text{-HDX-Uni}}$ for individual peptides (***Figure 6d***).

Indeed, peptides covering the extracellular inter-subunit interface (174–184 and 187–194 in TM4a-b; and 81–88 in TM2) showed the lowest $T_{50\text{-HDX-Uni}}$ values that ranged between 50.6–52.8°C and 40.6–42.1°C in EAAT1$_{CO}$ and EAAT1$_{COCO}$, respectively, demonstrating that unfolding of this region is the first structural change observed during thermal denaturation (***Figure 6e,f***). Notably, the excellent agreement between the $T_{50\text{-SEC}}$ obtained for the subunit dissociation and the $T_{50\text{-HDX-Uni}}$, as well as the $T_{50\text{-HDX-Bi}}$, show that the structural changes in TM4a-b, and possibly in TM2, are key rate-limiting steps that determine the thermal stability of the trimeric and functional state of the transporters. Interestingly, the $T_{50\text{-HDX-Uni}}$ values of peptides covering the TranD of EAAT1$_{CO}$ ranged between 52.8°C in the cytoplasmic end of TM3 to 56–58°C in HP1, TM7, and TM8, demonstrating that the regions involved in substrate and ion coordination unfold at later stages in the thermal denaturation pathway of the transporters.

## Discussion

### Thermal denaturation pathway

EAAT1$_{WT}$ looses its transport function after purification in detergent solutions and shows extensive unfolding of several transmembrane regions important for oligomerization and transport function, highlighting the importance of the plasma membrane in stabilizing folded transporters. In stark contrast, EAAT1$_{CO}$ and EAAT1$_{COCO}$ remain functional and natively folded in detergent solutions,

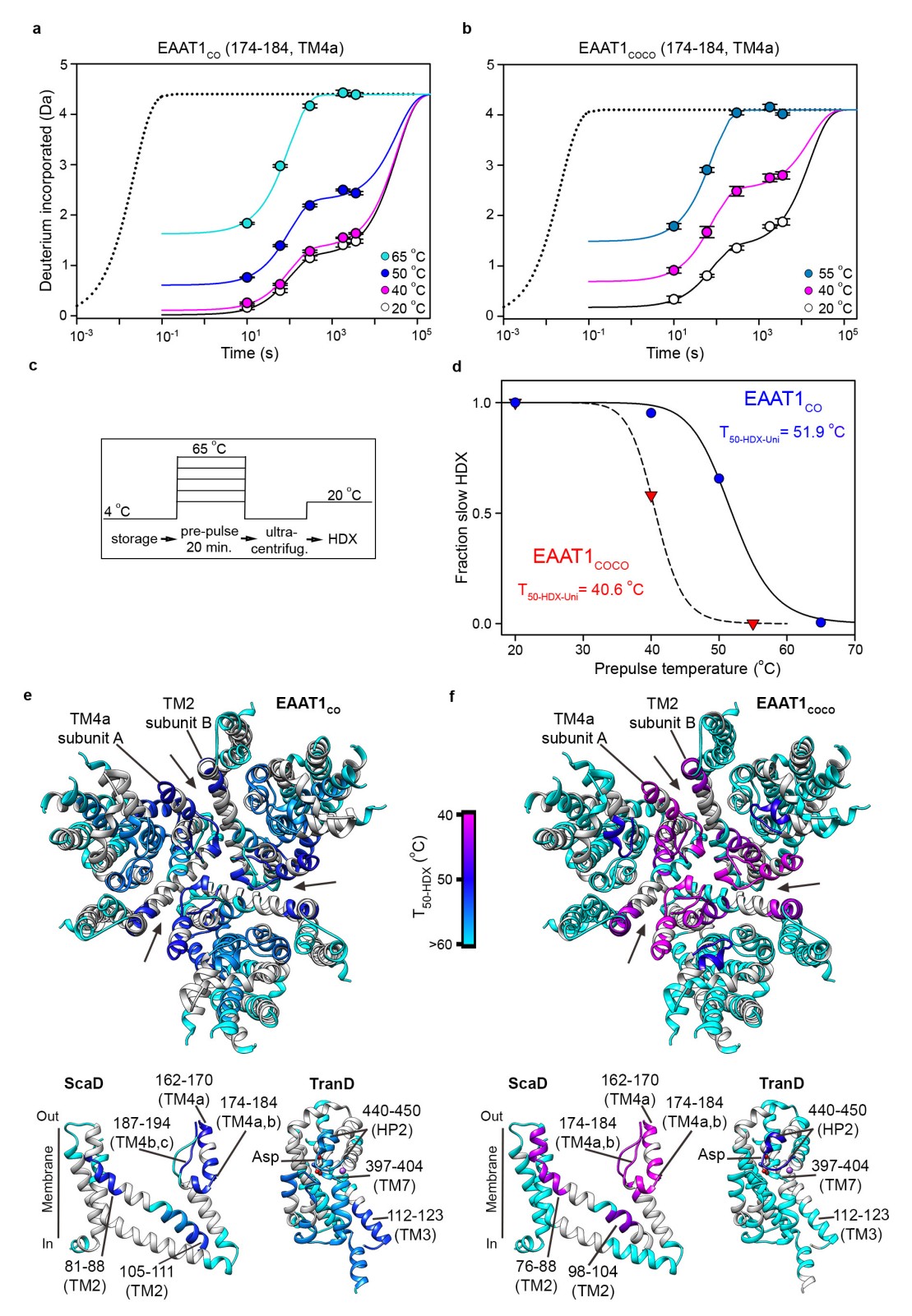

**Figure 6.** | Local thermal unfolding. (a-b) Deuterium uptake kinetics at different pre-pulse temperatures of example peptides containing residues 174–184 in EAAT1_CO (a), and EAAT1_COCO (b), respectively. Solid lines represent double-exponential fits to the data, and dotted lines the expected deuterium kinetics of the unfolded and solvent exposed peptide at 20°C (see methods). Plots in (a–b) depict an average of three independent experiments, and error bars represent s.e.m. (c), Depiction of the temperature protocol used to prepare the protein samples. (d), The fraction of the

*Figure 6 continued on next page*

*Figure 6 continued*

slow HDX component extracted from the double-exponential fit to the kinetic data in (a) and (b) corresponding to EAAT1$_{CO}$ (blue symbols) and EAAT1$_{COCO}$ (red symbols), respectively. Lines represent the fit of a Hill-like equation with T$_{50-HDX-Uni}$ values 51.9°C and 40.6°C in EAAT1$_{CO}$ and EAAT1$_{COCO}$, respectively, and H values of −18.7 and −20, respectively. (e–f) T$_{50-HDX-Uni}$ values, calculated as in (d), of EAAT1$_{CO}$ (e) and EAAT1$_{COCO}$ (f), respectively, are mapped into the structure of the EAAT1$_{CRYST}$ (PDB 5LLM) trimer viewed from the extracellular medium (upper panel), as well as the scaffold (ScaD) and the transport (TranD) domains viewed from the membrane (lower panel), respectively. These domains are depicted separately for clarity of display. The color code representing T$_{50-HDX-Uni}$ values is depicted in a scale bar between the trimers. Peptides that did not show any temperature-induced HDX changes are also labeled in cyan (T$_{50-HDX-Uni}$>60°C).

DOI: https://doi.org/10.7554/eLife.40110.016

The following figure supplement is available for figure 6:

**Figure supplement 1.** | Deuterium uptake kinetics in EAAT1$_{CO}$ and EAAT1$_{COCO}$ at different temperatures.

DOI: https://doi.org/10.7554/eLife.40110.017

showing that the consensus exchanges effectively compensate for the destabilizing effect of detergent micelles and the lack of lipid bilayer, while conserving the neurotransmitter transport function.

Interestingly, the two consensus designs show similar sequence of temperature-induced structural changes suggesting a common thermal denaturation pathway (*Figure 7*). Moreover, the deuterium uptake pattern of EAAT1$_{WT}$ resembles that of EAAT1$_{CO}$ after being denatured by a 65°C pre-pulse suggesting that the wild type transporter might also unfold through a similar pathway. In this pathway, the initial structural changes map to the inter-subunit interface formed between TM4a-b and TM2 from different subunits, respectively. However, we are not able to rule out potential changes in other regions of that interface, particularly at the level TM4c and TM5, due to lack of sequence coverage in our HDX-MS experiments. Notably, the excellent quantitative agreement between the T$_{50-HDX}$ values of interfacial peptides and the T$_{50-SEC}$ in both EAAT1$_{CO}$ and EAAT1$_{COCO}$, despite the ~10°C difference in stability, shows that unfolding of the inter-subunit interface is highly correlated to the loss of quaternary structure, and constitutes an important rate limiting step during the thermal denaturation of the transporters. Although, at present it is not possible to establish if the interfacial unfolding events are cause or effect of the subunit dissociation, they likely contribute to make such process irreversible.

In contrast, the TranD in the two consensus designs show no temporal correlation with the subunit dissociation and unfolds later in the pathway. Moreover, the TranD along with the substrate and

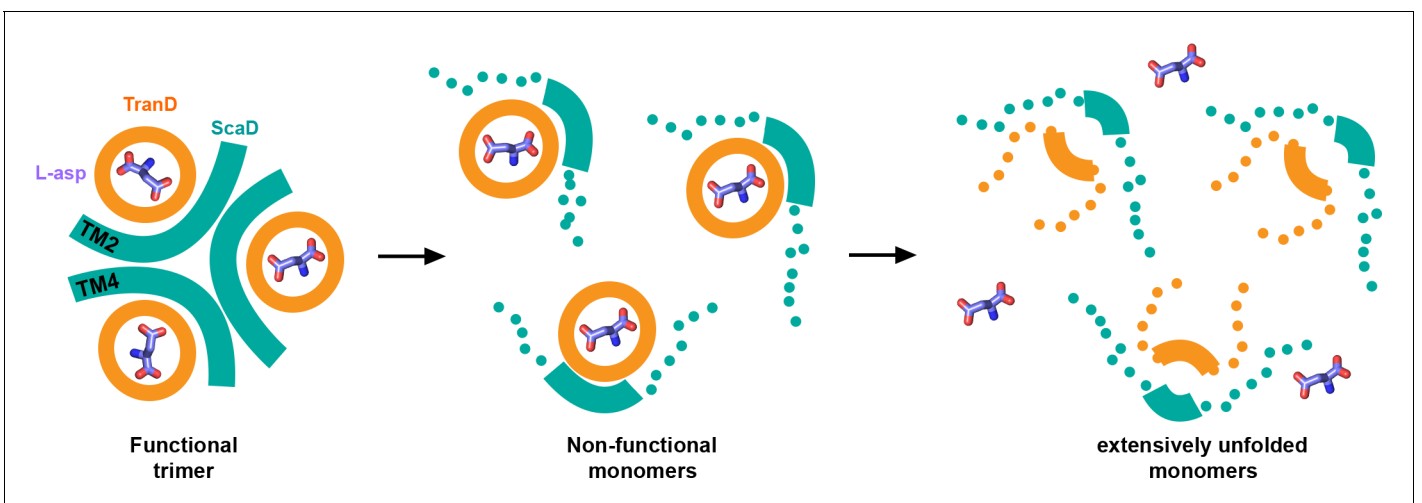

**Figure 7.** | Thermal unfolding pathway. Cartoon representation of the temperature-induced structural changes observed in the consensus designs. At the reference temperature the transporters are trimeric and have the substrate (sticks) bound (left panel). As temperature increases, the inter-subunit interface (cyan) between TM4 and TM2 unfolds and the subunits dissociate loosing their transport function (middle panel). Eventually, at high temperatures, the transport domains unfold and loose the ability to bind the substrate (right panel). Thick lines represent natively folded regions of the transporters, and dotted lines unfold ones.

DOI: https://doi.org/10.7554/eLife.40110.018

ions binding sites in EAAT1$_{CO}$ and EAAT$_{COCO}$ have overall similar thermal stabilities, suggesting that the additional mutations present in the TranD of EAAT1$_{CO}$ (19 mutations), compared to EAAT1$_{COCO}$, do not contribute strongly to the transporter stability.

The important question arises of what unfolding events cause the functional inactivation of the transporters? The transport experiments done with EAAT1$_{COCO}$ pre-heated at 55°C showed that the monomeric transporters are not functional. Yet, HDX analysis shows that under the same conditions the TranD domain is natively folded. Therefore one likely possibility is that despite the functional independence of the glutamate transporter subunits, they can only function in the context of the trimer to avoid that the substrate binding site and dynamic regions important for transport (e.g. HP2) become buried in the bilayer, or constrained by lipid or detergent molecules. Interestingly, we observed decreased deuterium uptake at the level of HP2 upon dissociation of the subunits in both EAAT1$_{CO}$ and EAAT1$_{COCO}$, consistent with this hypothesis.

Our discovery of early unfolding events at the trimeric interface that determine the lifetime of the transporters opens the interesting possibility to target this region by mutagenesis, or even small-molecule compounds that could extend the half-life of the transporters and possibly act as the long-sought glutamate transport activators.

## Consensus-based approaches to thermal stability

Using consensus-based mutagenesis approaches, we engineered increased kinetic stability to a human oligomeric and multidomain integral membrane protein of ~180 kDa (EAAT1), and trapped it in folded and functional states for biophysical analysis. Interestingly, consensus amino acids were identified from an alignment of two divergent branches of animal SLC1 proteins: sodium- and potassium-dependent glutamate transporters (EAAT1-5), as well sodium-dependent neutral amino acid transporters (ASCT1-2). The fact that our consensus designs show conserved sodium- and potassium-dependent glutamate transport function might simply reflect the higher number of sequences with such function in the alignment. More intriguing is the fact that the alignment includes exclusively SLC1 sequences from cold- and warm-blooded animals, and lacks those from thermophilic organisms. Pioneering work on soluble proteins has shown that individual consensus mutations have a higher chance to be stabilizing than random ones (*Steipe, 2004*; *Steipe et al., 1994*; *Nikolova et al., 1998*). In addition, protein evolution is driven by random mutagenesis to achieve specialized functions, and some degree of protein stability loss is expected during specialization, despite the acquisition of compensatory stabilizing mutations (*Tokuriki and Tawfik, 2009*). Hence, a sequence containing the most frequent amino acids among structural homologs, that is consensus amino acids, will on average filter out some of the non-stabilizing mutations acquired during the specialization of the individuals, and confer higher stability to the fold. Several expectations follow from this line of thought: first, the impact of consensus mutagenesis in protein stability should be smaller in protein families that have high selective pressure to retain stability, in other words, protein families in which stability is part of the specialized function. Although systematic studies will be needed to test this, the loss of stability observed in a full-consensus sequence of a highly thermostable soluble protein supports the above expectation (*Dai et al., 2007*). Second, less-specialized ancestral proteins are expected to be more stable than the extant specialized structural homologs. Indeed, gain of stability has been observed in several laboratory resurrections of ancestral soluble proteins (*Gaucher et al., 2008*; *Perez-Jimenez et al., 2011*; *Risso et al., 2013*; *Wheeler et al., 2016*). Third, in multi-domain proteins, conserved domains are expected to be more stable than divergent ones, since they contain more consensus residues. This is in fact what we observed in EAAT1, in which the conserved transport domain is more thermostable than the divergent scaffold domain.

In our second consensus design, EAAT1$_{COCO}$, we tested the possibility that inter-residue contacts predicted by amino acid covariance are important for stability, and show that restricting the consensus mutations to highly covariant residue pairs yields folded and functional transporters. It is worth noting that in this COCO design, two different sequence alignments were used, one to determine the positions in the sequence to be mutated (curated alignment of the PFAM SLC1 family), and one to determine the identity of the consensus amino acids (alignment of animal SLC1 proteins). This is likely key to the success of the approach, since consensus amino acids from very divergent sequences, like those in the PFAM alignment that is dominated by prokaryotic SLC1 homologs, could disrupt important conserved interactions for the stability of the human transporters. In agreement to this, consensus mutations calculated from curated PFAM alignments of soluble enzymes, at strongly

correlated positions, have shown detrimental effects on stability (*Sullivan et al., 2011*; *Sullivan et al., 2012*; *Durani and Magliery, 2013*).

The comparison between EAAT1$_{CO}$ and EAAT1$_{COCO}$ shows that the former is more stable than the latter. Clearly, one caveat in the design of EAAT1$_{COCO}$ is that the PFAM alignment has a strong phylogenetic bias for prokaryotes, and could include evolutionary coupled positions that are not important or even detrimental for the stability of human transporters. An additional explanation is that the consensus-covariance approach focuses on amino acid contacts and fails to capture other interactions like protein-lipid or protein-detergent that are important for membrane protein stability.

The consensus-based approaches presented here could be generalized to other human membrane protein families to obtain stable and functional proteins for structural, as well as biophysical analyses. An advantage over current methods like alanine scanning and direct evolution is that the consensus-based approaches reduce the number of constructs to be screened by nearly two orders of magnitude, since only a few synthetic genes need to be tested. However, a potential disadvantage is the larger number of mutations in the consensus designs compared to thermally stabilized animal and human membrane proteins obtained by the above-mentioned methods (*Zhou and Bowie, 2000*; *Magnani et al., 2008*; *Sarkar et al., 2008*; *Coleman et al., 2016*). Further work will be required to understand the impact of the depth and phylogenetic bias in the alignments used for the consensus designs, and to maximize protein stability while minimizing the number of consensus mutations.

# Materials and methods

**Key resources table**

| Reagent type (species) or resource | Designation | Source or reference | Identifiers | Additional information |
|---|---|---|---|---|
| Gene (*Homo sapiens*) | EAAT1$_{WT}$ | DNA2.0 synthetic | Uniprot P43003-1 | Codon optimized Homo sapiens |
| Cell line (*Homo sapiens*) | HEK293F | Thermo Fisher | R79007 | negative for mycoplasma |
| Transfected construct (*Homo sapiens*) | EAAT1$_{CO}$ | This paper | | |
| Transfected construct (*Homo sapiens*) | EAAT1$_{COCO}$ | This paper | | |
| Transfected construct (*Homo sapiens*) | EAAT1 Sca$_{CO}$-TranD$_{COCO}$ | This paper | | |
| Recombinant DNA reagent | pCDNA3.1 | Invitrogen, doi: 10.1038/nature22064 | | |
| Software, algorithm | FELIX 4.1.2 | Photon Technology International/Horiba | | |
| Software, algorithm | DynamX 3.0 | Waters | | |
| Software, algorithm | HX-Express2 | http://www.hxms.com/HXExpress | | |
| Software, algorithm | Sigma Plot 12 | Sysat Inc. | | |
| Software, algorithm | Excel | Microsoft | | |

## Consensus designs

Consensus amino acids were calculated using JALVIEW (*Waterhouse et al., 2009*) from 113 aligned sequences using Muscle (*Edgar, 2004*) of vertebrate homologs of the seven human SLC1 family members (*Gesemann et al., 2010*) (SLC1A1-7) (available in FASTA format in *Supplementary file 1*). A consensus amino acid at any position was defined as the most frequent amino acid, given that it shows a total frequency of appearance $>\sim 20\%$, and this frequency was $\sim 10\%$ higher than that of the corresponding amino acid in human EAAT1. To generate EAAT1$_{CO}$, we simultaneously introduced all consensus amino acid exchanges defined as above in regions where secondary structure was

predicted, based on the structures of GltPh (PDB 1XFH, 3KBC). To generate EAAT1$_{COCO}$, we selected a subset of the consensus exchanges introduced in EAAT1$_{CO}$ at positions in the EAAT1 sequence that show strong amino acid covariance. Amino acid covariance was calculated with the EVcouplings server (http://evfold.org/evfold-web/evfold.do) (*Marks et al., 2011*) using the pseudo-likelihood maximization approach. The input for the calculations was a manually curated alignment of the PFAM 'sodium:dicarboxylate symporter' family (PF00375) with 4275 sequences of SLC1 homologs (available in FASTA format in *Supplementary file 2*), including prokaryotic and eukaryotic organisms, and covering at least 70% of the human EAAT1 amino acid sequence (P43003-1 UniProt). From this analysis, the top covariance scored residue pairs predicted physical contacts (<6 Å apart) in the outward- and inward-facing state crystal structures of the prokaryotic homolog GltPh (PDB 1XFH, 3KBC) at a success rate of ~ 90%. Then, after removing false negative contacts using the above-mentioned GltPh trimeric structures, we focused the consensus amino acid exchanges to the top 100 amino acid pairs with the highest amino acid covariance scores to generate EAAT1$_{COCO}$. It is important to emphasize here, that the covariance analysis was used only to select the subset of positions in the EAAT1 sequence where to preform the consensus amino acid exchanges. However, these exchanges were identical to those introduced in EAAT1$_{CO}$, and calculated from the alignment of vertebrate SLC1 homologs. Hence, neither of the consensus designs carries mutations from distant prokaryotic homologs that dominate the PFAM alignment.

## Protein expression, purification and size-exclusion chromatography

Synthetic genes codifying for EAAT1$_{WT}$, EAAT1$_{CO}$, and EAAT1$_{COCO}$ were codon optimized for expression in human cell-lines and purchased (DNA2.0 and Invitrogen). All genes were cloned into pcDNA3.1(+) (Invitrogen) with N-terminal Strep-tag II affinity tag followed by eGFP and PreScission protease cleavage site, and expressed in HEK293F cells (Thermo Fisher, mycoplasma test negative, and no authentication was attempted) grown in Excell293 medium (Sigma) and supplemented with 4 mM L-glutamine (Sigma) and 5 μg ml$^{-1}$ Phenol red (Sigma-Aldrich) to densities of $2.5 \times 10^6$ cells ml$^{-1}$. Cells were transiently transfected in Freestyle293 medium (Invitrogen) using 9 μg ml$^{-1}$ polyethylenimine (PEI) (Polysciences) and 3 μg ml$^{-1}$ of plasmid DNA, at a density of $2.5 \times 10^6$ cells ml$^{-1}$, diluted with an equivalent volume of Excell293 6 hr after transfection, and treated with 2.2 mM valproic acid (Sigma) 12 hr after dilution of the cultures. Cells were collected at around 48 hr after transfection in 50 mM HEPES/Tris-base, pH 7.4, buffer supplemented with 1 mM L-Asp, 1 mM EDTA, 1 mM PMSF, 1 mM TCEP, and 1:200 (v/v) dilution of mammalian protease inhibitor cocktail (Sigma), and disrupted in an cell homogenizer (EmulsiFlex-C5, Avestin) after three runs at approximately 125,000 kPa. The resulting homogenate was clarified by centrifugation (4,500 g, 0.5 hr) and the crude membranes were collected by ultracentrifugation (186,000 g, 1.5 hr). Membranes were washed once with the above-mentioned buffer and finally homogenized with a douncer in a buffer containing 50 mM HEPES/Tris-base, pH 7.4, 200 mM NaCl, 1 mM L-Asp, 1 mM EDTA, 1 mM TCEP, and 5% glycerol, snap-frozen in liquid N$_2$ and stored at − 80°C at 0.25 g membranes ml$^{-1}$. Membrane solubilization was done by thawing out and supplementing the membrane homogenate with 2% sucrose monododecanoate (DDS) and 0.4% cholesterol hemi-succinate (CHS). After 1 hr incubation, the insoluble material was removed by ultracentrifugation (186,000 g for 1 hr), and Strep-Tactin sepharose resin (GE Healthcare) was added to the supernatant and rotated for 2 hr. Resin was washed with 25 column volumes of 50 mM HEPES/Tris-base, pH 7.4, 200 mM NaCl, 1 mM L-Asp, 0.5 mM TCEP, 5% glycerol, 0.0632% DDS and 0.0126% CHS, and the protein was eluted with the same buffer supplemented with 2.5 mM L-desthiobiotin. The eluted eGFP-transporter fusion was concentrated to 500 μl using 100 kDa cutoff membranes (Millipore), ultra-centrifuged (86,900 g, 20 min), and applied to a Superose 6 10/300 gel filtration column (GE Healthcare) equilibrated with 50 mM HEPES/Tris-base, pH 7.4, 200 mM NaCl, 1 mM L-Asp, 0.5 mM TCEP, 5% glycerol, 0.0632% DDS and 0.01264% CHS.

To study the thermal stability of the different constructs, the purified protein at a concentration of 0.2 mg ml$^{-1}$ in 50 mM HEPES/Tris-base, pH 7.4, 200 mM NaCl, 1 mM L-Asp, 0.5 mM TCEP, 5% glycerol, 0.0632% DDS and 0.01264% CHS was diluted 1:20 in the same buffer, and incubated for 20 min at a single temperature in a C1000 Touch Thermal Cycler (BioRad). 100 μl of sample were ultra-centrifuged (86,900 g, 20 min) to clear the solution, and loaded in a 96-well microplate (Greiner Bio-One), for injection in an SRT-SEC-500 column (Sepax). The elution profiles were detected with a QuantaMaster 400 fluorometer (PTI) set at $\lambda_{ex}$ = 470 nm and $\lambda_{em}$ = 514 nm using a flow-through cell

(Starna). The area under the full chromatographic profile was integrated numerically and remained relatively constant at all temperatures studied (empty symbols *Figure 3d–f*). The fractional area corresponding to the trimeric transporters was calculated as the fraction of the area under the high molecular-weight peak (elution at ~3 ml) over the total area of the chromatogram. The values of the fractional areas corresponding to the trimeric transporters were fitted to a Hill equation of the form:

Norm. area = $1/(1+(T_{50\text{-SEC}}/T)^H)$ in which $T_{50\text{-SEC}}$ is the temperature at which half of the transporters are in the trimeric state, and H the Hill coefficient.

## Radioactive substrate transport

Unilamellar liposomes were made at 9:1 molar ratio of 1-palmitoyl-2-oleoyl-sn-glycero-3-phosphocholine (Avanti Polar Lipids) and CHS, in a buffer containing 50 mM HEPES/Tris-base, pH 7.4, 200 mM NaCl and 1 mM L-Asp. To reconstitute the protein, liposomes were first mixed with DDS at a 1:2 (w/w) lipid-to-detergent ratio for 1 hr, and then the purified transporters were added at a 1:40 (w/w) protein-to-lipid ratio. Detergent removal was done at 4°C using SM-2 biobeads (BioRad) at 100 mg ml$^{-1}$. The internal solution of the liposomes was exchanged using 10 freeze-thaw cycles in 50 mM HEPES/Tris-base, pH 7.4, and 200 mM KCl, unless indicated otherwise. After extrusion through 400 nm polycarbonate membranes (Avanti Polar Lipids), the proteoliposomes were concentrated by ultracentrifugation (150,000 g for 30 min at 4°C) and resuspended at 20 mg lipids ml$^{-1}$, for immediate use. Substrate transport was assayed at 37°C. The uptake reaction was initiated by diluting the proteoliposomes tenfold into a buffer containing 50 mM HEPES/Tris-base, pH 7.4, 200 mM NaCl, 50 μM L-glutamate, 5 μM [$^{14}$C] L-glutamate (PerkinElmer), and 2.5% glycerol. After 30 min, 200 μl aliquots were diluted fivefold into ice-cold quench buffer (50 mM HEPES/Tris-base, pH 7.4, 200 mM ChCl, and 2.5% glycerol), followed by immediate filtration and wash on nitrocellulose 0.22 μm filters (Millipore). Radioactivity was quantified by liquid scintillation using a Tri-Carb 3110TR counter (PerkinElmer). Background radioactivity was estimated from protein-free liposomes, and subtracted from the uptake data. The values were normalized by the amount of reconstituted protein, estimated by the level of GFP fluorescence measured in an Infinite M1000Pro microplate reader (Tecan). To titrate the rate of L-glutamate transport, proteoliposomes were assayed in the presence of 0, 5, 50 or 200 μM L-glutamate supplemented with 1, 5, 5, or 5 μM [$^{14}$C] L-glutamate, respectively. At each substrate concentration, the initial rate of transport was calculated by a linear fit to 120 s and 180 s uptake measurements with origin fixed at zero. For the cell-based transport uptake, cells were collected 36 hr after transfection, and washed three times and resuspended at a density of 50 × 10$^6$ cells ml$^{-1}$ in 11 mM HEPES/Tris-base, pH 7.4, 140 mM ChCl, 4.7 mM KCl, 2.5 mM CaCl$_2$, 1.2 mM MgCl$_2$, and 10 mM D-glucose, for immediate use. The uptake assay was performed similarly to the one described for the proteoliposomes, but using a reaction buffer containing 11 mM HEPES/Tris-base, pH 7.4, 140 mM NaCl, 4.7 mM KCl, 2.5 mM CaCl$_2$, 1.2 mM MgCl$_2$, 10 mM D-glucose, 50 μM L-glutamate, and 5 μM [$^{14}$C] L-glutamate, and 0.8 μm nitrocellulose filters. Background radioactivity was estimated from cells transfected with empty vector, and subtracted from the uptake data.

## Hydrogen-deuterium exchange mass spectrometry

HDX-MS experiments were performed with transporters purified as described above, but after removal of eGFP following elution. The eluted eGFP-transporter fusion was digested with PreScission protease overnight at 4°C, concentrated to 100 μL using 100 kDa cutoff membranes (Millipore), ultra-centrifuged for 20 min at 86,900 g, and finally applied to a Superose 6 5/150 gel filtration column equilibrated with 50 mM HEPES/Tris-base, 200 mM NaCl, pH 7.4, 1 mM L-Asp, 0.5 mM TCEP, 0.0632% DDS, 0.01264% CHS and 5% glycerol. To evaluate and compare the stability of the EAAT1 consensus designs by HDX-MS, the purified proteins were heated at a single pre-pulse temperature for 20 min in a C1000 Touch Thermal Cycler (BioRad), ultra-centrifuged (86,900 g, 20 min) to clear the solution, and further equilibrated for 1 hr at 20°C before deuterium labeling.

Deuterium exchange was initiated by adding 40 μL of D$_2$O buffer (50 mM HEPES, pD 7.4, 200 mM NaCl, 1 mM L-Asp, 5% glycerol, 0.0632% DDS, 0.01264% CHS, 0.5 mM TCEP) to 10 μL of EAAT1 protein solution at ~5 μM (monomer concentration). Continuous labeling was performed at 20°C for t = 10, 60, 300, 1800 and 3,600 s. Aliquotes of 10 μL (10 pmoles) were withdrawn at each time point and quenched upon mixing with 50 μL of ice-cold quench buffer (0.75% formic acid, 5%

glycerol) to reduce the pH to 2.5. Quenched samples were immediately snap-frozen in liquid nitrogen and stored at −80°C. Undeuterated controls were treated using an identical procedure. Experiments were performed in triplicate for each time point and condition.

8.3 pmol of quenched protein samples were injected into a nanoACQUITY UPLC HDX system (Waters) maintained at 0°C. Samples were on-line digested for 2 min at 100 µL/min and 20°C using an in-house packed immobilized pepsin cartridge (2.0 × 20 mm, 63 µL bed volume). The resulting peptides were trapped, concentrated and desalted onto a C18 Trap column (VanGuard BEH 1.7 µm, 2.1 × 5 mm, Waters) at a flow rate of 100 µL/min of 0.15% formic acid, and then separated in 8 min by a linear gradient of acetonitrile from 5% to 30% at 40 µL/min using an ACQUITY UPLC$^{TM}$ BEH C18 analytical column (1.7 µm, 1 × 100 mm, Waters). After each run, the pepsin cartridge was cleaned with two consecutive washes of 1% formic acid, 5% acetonitrile, 1.5 M guanidinium chloride, pH 2.5. Blank injections were performed between each run to confirm the absence of carry-over.

Mass spectra were acquired in positive and resolution mode on a Synapt G2-Si HDMS mass spectrometer (Waters) equipped with a standard ESI source and lock-spray correction (Glu-Fibrinogen peptide at 100 fmol/µL in 50% acetonitrile). Peptide maps were generated with the Protein LynX Global Server 3.0 (Waters) using MS$^E$ data collected on undeuterated controls. Each fragmentation spectrum was manually inspected to confirm the assignment. Deuterium uptake values were extracted using DynamX 3.0 (Waters). Only one unique charge state was considered per peptide and no adjustment was made for back-exchange; HDX-MS results are therefore reported as relative deuterium uptake values. Statistical analysis was performed with MEMHDX using a False Discovery Rate of 1% (*Hourdel et al., 2016*).

For each EAAT1 protein, bimodal *m/z* envelopes were first detected manually and corroborated with the binomial fitting approach implemented in HX-Express2 (*Guttman et al., 2013*). Double-Gaussian fitting to averaged *m/z* envelopes was used to determine the temperature-induced fractional changes of the low and high-*m/z* peptides' components using Sigma Plot 12.0 (Systat Software Inc.). The fractional amplitude of the low m/z component as a function of the pre-pulse temperature was fitted to a Hill equation of the form (*Figure 5e,f*):

Fract. Amplitude = $1/(1+(T_{50\text{-HDX-Bi}}/T)^H)$

Where H is the Hill coefficient (constrained to > −20), T is the pre-pulse temperature, and $T_{50\text{-HDX-Uni}}$ is the temperature at which the fraction of the low m/z component equals 0.5.

## HDX kinetic analysis

Deuterium uptake values were calculated from the centroid of unimodal isotopic m/z envelopes measured at 20°C in the absence of temperature pre-pulses. These values were fitted to a double exponential function of the form:

Deuterium Uptake= $A_0 + A_1*[1\text{-exp}(k_1*t)] + A_2*[1\text{-exp}(k_2*t)]$

Where $A_0$ is the amplitude of the initial deuterium uptake burst determined by uptake at 10 s, the shortest time point measured. $A_1$ and $A_2$ are the amplitudes, and $k_1$ and $k_2$ the rate constants of the intermediate and slow uptake components, respectively. In order to compare the deuterium uptake kinetics of EAAT1$_{CO}$, EAAT1$_{COCO}$, and EAAT1$_{WT}$, we constrained the maximal amplitude of deuterium uptake ($A_0+A_1+A_2$) in the fits to the maximal value observed in the EAAT1$_{WT}$ peptides, which showed clear saturating values at the longest time points measured (1800 and 3,600 s) in most cases (*Figure 2*, *Figure 2—figure supplement 4*). Using this constrain, dynamic peptides showed deuterium uptake kinetics with $k_1$ values of ~$10^{-1}$–$10^{-2}$ s$^{-1}$ and $k_2$ values of <$10^{-3}$ s$^{-1}$, respectively, in both EAAT1$_{CO}$ and EAAT1$_{COCO}$. For comparison the expected deuterium kinetics of fully unprotected peptides was estimated using an empirical overall HDX rate calculated with the SPHERE server (http://landing.foxchase.org/research/labs/roder/sphere/sphere.html). This rate was the sum of the individual exchange rates from the amide hydrogen atoms available for exchange in a peptide. The first amide in the peptide was systematically excluded.

The deuterium uptake values of EAAT1$_{CO}$ and EAAT1$_{COCO}$ samples pre-heated at different temperatures were also calculated from the centroids of unimodal isotopic envelopes and fitted to the same double exponential equation. However, in this analysis we used a global fitting protocol for the data of a given peptide at all pre-pulse temperatures measured, in which $k_1$ and $k_2$ parameters were shared for the data points at all temperatures. Moreover, the global fits were constrained to a maximal deuterium uptake value ($A_0+A_1+A_2$) corresponding to the saturating values observed for a given peptide at the highest temperature measured, 65°C and 55°C for EAAT1$_{CO}$ and EAAT1$_{COCO}$,

respectively. Then, the fractional amplitude of the slow deuterium uptake component ($A_2/A_0+A_1+A_2$) was computed for each pre-pulse temperature, and normalized to the value measured at the reference temperature (20°C). The normalized fractional amplitude of the slow component as a function of temperature, for a given peptide, was finally fitted to a Hill equation of the form (*Figure 6d*, *Figure 6—figure supplement 1*):

Norm. Frac. Amplitude = $1/(1+(T_{50\text{-HDX-Uni}}/T)^H)$

Where H is the Hill coefficient (constrained to $> -20$), T is the pre-pulse temperature, and $T_{50\text{-HDX-Uni}}$ is the temperature at which the fractional amplitude of the slow component equals 0.5.

## Acknowledgements

We thank Olga Boudker for discussion on consensus mutagenesis; Petya V Krasteva for comments on the manuscript. The work was funded by the ERC Starting grant 309657 (NR). Further support from G5 Institut Pasteur funds (NR), CACSICE grant (ANR-11-EQPX-008), and CNRS UMR3528 (NR, JC-R.) is acknowledged.

## Additional information

### Competing interests

Nicolas Reyes: Is inventor on PCT/FR2018/050371 describing the use of consensus mutagenesis to modify protein thermal stability. The other authors declare that no competing interests exist.

### Funding

| Funder | Grant reference number | Author |
|---|---|---|
| H2020 Excellent Science | ERC Starting grant 309657 | Nicolas Reyes |
| Centre National de la Recherche Scientifique | UMR 3528 | Julia Chamot-Rooke Nicolas Reyes |
| Agence Nationale de la Recherche | CACSICE grant ANR-11-EQPX-008 | Julia Chamot-Rooke Nicolas Reyes |

The funders had no role in study design, data collection and interpretation, or the decision to submit the work for publication.

### Author contributions

Erica Cirri, Formal analysis, Investigation, Visualization, Methodology, Writing—original draft, Writing—review and editing; Sébastien Brier, Formal analysis, Investigation, Visualization, Writing—original draft, Writing—review and editing; Reda Assal, Juan Carlos Canul-Tec, Investigation, Methodology; Julia Chamot-Rooke, Supervision, Funding acquisition, Writing—original draft; Nicolas Reyes, Conceptualization, Formal analysis, Supervision, Funding acquisition, Visualization, Methodology, Writing—original draft, Writing—review and editing

### Author ORCIDs

Sébastien Brier http://orcid.org/0000-0003-1758-8237
Nicolas Reyes http://orcid.org/0000-0001-6618-8307

### Decision letter and Author response

Decision letter https://doi.org/10.7554/eLife.40110.023
Author response https://doi.org/10.7554/eLife.40110.024

## Additional files

### Supplementary files

• Supplementary file 1. Alignment animal SLC1 homologs.
DOI: https://doi.org/10.7554/eLife.40110.019

- Supplementary file 2. Alignment PFAM curated SLC1.
DOI: https://doi.org/10.7554/eLife.40110.020

- Transparent reporting form
DOI: https://doi.org/10.7554/eLife.40110.021

All data generated or analysed during this study are included in the manuscript. Supporting files including the amino acid sequence alignments used in this study are also provided

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
