## [Decision Letter]

Thank you for submitting your article "Consensus designs and thermal stability determinants of a human glutamate transporter" for consideration by *eLife*. Your article has been reviewed by two peer reviewers, and the evaluation has been overseen by José D. Faraldo-Gómez as Reviewing Editor and Richard Aldrich as the Senior Editor. The following individuals involved in review of your submission have agreed to reveal their identity: Christopher G Tate (Reviewer #1).

The reviewers have discussed the reviews with one another and the Reviewing Editor has drafted this decision to help you prepare a revised submission.

Summary:

Cirri et al. have performed detailed analyses on the thermal stability of a glutamate transporter, in the wild-type form as well as for thermally-stable variants identified through a computational sequence-based procedure. The manuscript is important in two ways. First, it details an approach that, if shown to be generalizable to other membrane proteins, would represent an important advancement in the structural biology field. Increasing the kinetic stability of membrane proteins without loss of function is a goal pursued by many, as it would facilitate the discovery of new architectures or new conformational states while ensuring the structural data is mechanistically relevant. The second key finding is that interactions at the interface between the so-called scaffold and transport domains are important determinants of the kinetic stability of this protein. It is plausible that this finding is transferable to other transporters whose functional mechanism entails relative motions between distinct structural domains.

Essential revisions:

1) The reviewers ask that the comparison between the rates in the text (subsection “Structural comparison between EAAT1 wild type and consensus designs”) and the data in Figure 2 be made clearer; that the latter are expressed in terms of (the reciprocal) time in the x-axis can be confusing. Also, the reviewers ask that the authors justify the double exponential fits to some of these data. Figure 2A for example certainly does not define a double exponential. The authors ought to specify what the values of the degrees of freedom are for these fits and/or if some parameters in the fitting equations are fixed, as well as the rationale to do so.

2) The authors must clarify whether conservation of mass was calculated for the gel filtration runs. The reviewers are surprised that the monomer (native or non-native) does not just stick to the column and never come off (slowing the flow rate and raising the pressure in the meantime). While some loss of this species would not change the overall findings of this manuscript, it could influence the population outcomes.

3) The authors should discuss the extent to which the success of the proposed method depends on the "manual curation" of sequences, and explain in detail the criteria used in this particular study as well as its rationale.

---

## [Author Response]

Essential revisions:1) The reviewers ask that the comparison between the rates in the text (subsection “Structural comparison between EAAT1 wild type and consensus designs”) and the data in Figure 2 be made clearer; that the latter are expressed in terms of (the reciprocal) time in the x-axis can be confusing.

To improve clarity, we revised the text and describe deuterium uptake time-courses in terms of Time only (not rates). The x-axis units in Figure 2A-C and Figure 2—figure supplement 5 are also seconds. The revised text reads: “The deuterium uptake time course of most peptides covering structured regions of the TranD and ScaD based on the structure of EAAT1_CRYST_ showed slow deuterium incorporation that starts after ≥ 10 s, or even lacked any incorporation for up to 1 hour (Figure 2A-C and Figure 2—figure supplement 5)”

We also edited the initial paragraph in the Results section “Rate-limiting structural changes of thermal unfolding” in the same way for consistency. It now reads: “Briefly, the HDX kinetics of any detected peptide was well described by three components: an initial burst determined by uptake measured at 10 s (the first time point in the experiments); an intermediate component determined by uptake within 10^1^-10^3^ s (where most of our experimental measurements were done); and a slow component with uptake at longer times > 10^3^ s.”

Also, the reviewers ask that the authors justify the double exponential fits to some of these data. Figure 2A for example certainly does not define a double exponential. The authors ought to specify what the values of the degrees of freedom are for these fits and/or if some parameters in the fitting equations are fixed, as well as the rationale to do so.

In order to compare the deuterium uptake kinetics of the three constructs, we constrained the saturating uptake level to that of the WT peptides at the longest time point (3600 s), because it is the closest experimental value we obtained to maximal uptake for a given peptide. This is explained in the Materials and methods section “HDX kinetic analysis”: “we constrained the maximal amplitude of deuterium uptake (A_0_+A_1_+A_2_) in the fits to the maximal value observed in the EAAT1_WT_ peptides, which showed clear saturating values at the longest time points measured (1,800 and 3,600 s) in most cases (Figure 2 and Figure 2—figure supplement 4)”. We also refer to this section in the legend of Figure 2 for clarity.

Moreover, for most protected peptides in the consensus designs like 374-389 (Figure 2B and Figure 2—figure supplement 4), it is necessary to use a double exponential equation to fit the data and reach the maximal uptake value. For consistency, we used the double exponential fitting for all peptides, despite the fact that in a few examples like the one in Figure 2A, a single exponential could also fit the data. In any case, the main claim about uptake rates that we put forward in this section of the manuscript is independent of the fitting protocol: i.e. deuterium uptake of peptides in folded regions of the transporters, as predicted by the structure of EAAT1cryst, is orders of magnitude slower than expected for unfolded peptides of the same sequence and slower than observed experimentally for several peptides in EAAT1WT (e.g. peptide 397-404 Figure 2A or 374-389 Figure 2B).

2) The authors must clarify whether conservation of mass was calculated for the gel filtration runs. The reviewers are surprised that the monomer (native or non-native) does not just stick to the column and never come off (slowing the flow rate and raising the pressure in the meantime). While some loss of this species would not change the overall findings of this manuscript, it could influence the population outcomes.

Conservation of mass was calculated by numeric integration of the full chromatographic profiles. These so-called “total” areas obtained at different temperatures were then normalized to that at 4 °C for comparison across constructs, and are plotted as empty symbols in Figure 3D-F. We edited the legend of Figure 3, as well as the Materials and methods section “Protein expression, purification and size-exclusion chromatography” to improve clarity on this point.

As opposed to the area under the trimeric peak (solid symbols Figure 3D-F), the total areas are not temperature dependent, demonstrating protein mass conservation during pre-heating and ultracentrifugation. In addition, we do not observe any “clogging” signs in the SEC runs, like the ones mentioned by the Referee. Hence, we conclude that the consensus monomers (natively or partly folded) are soluble in detergent solutions under the reported conditions.

3) The authors should discuss the extent to which the success of the proposed method depends on the "manual curation" of sequences, and explain in detail the criteria used in this particular study as well as its rationale.

We don’t believe that manual curation of sequences plays an important role in the successful application of the method. First, there was no “manual curation” of sequences in the design of EAAT1CO. Second, the manual curation involved in the design of EAAT1COCO was done over the PFAM sequences to remove those that cover < 70% of the amino acid length of the target (EAAT1). This is important for the calculation of the amino acid covariance, and it can be done with the EVcouplings server as well, with similar results. We just noted a slightly lower rate of false-positives between co-evolved pairs and physical contacts when using “manually curated” sequences over “server-curated” sequence, but we do not think that it would have an impact on the outcome of the method. We added the following sentences in the Materials and methods section “Consensus designs” to reflect this: “Automated curation of PFAM sequences using the EVcouplings server yielded similar results”.